# AUTOENCODER-BASED HYBRID REPLAY FOR CLASS-INCREMENTAL LEARNING

## ABSTRACT

In class-incremental learning (CIL), effective incremental learning strategies are essential to mitigate task confusion and catastrophic forgetting, especially as the number of tasks $t$ increases. Current exemplar replay strategies impose $\mathcal{O}(t)$ memory/compute complexities. We propose an autoencoder-based hybrid replay (AHR) strategy that leverages our new hybrid autoencoder (HAE) to function as a compressor to alleviate the requirement for large memory, achieving $\mathcal{O}(0.1t)$ at the worst case with the computing complexity of $\mathcal{O}(t)$ while accomplishing state-of-the-art performance. The decoder later recovers the exemplar data stored in the latent space, rather than in raw format. Additionally, HAE is designed for both discriminative and generative modeling, enabling classification and replay capabilities, respectively. HAE adopts the charged particle system energy minimization equations and repulsive force algorithm for the incremental embedding and distribution of new class centroids in its latent space. Our results demonstrate that AHR consistently outperforms recent baselines across multiple benchmarks while operating with the same memory/compute budgets.

## 1 INTRODUCTION

Incremental learning addresses the challenge of learning from an upcoming stream of data with a changing distribution (Parisi et al., 2019; De Lange et al., 2021; Hadsell et al., 2020). To study incremental learning, two common scenarios are often considered: task-incremental learning (TIL) and class-incremental learning (CIL). CIL at the test phase requires jointly discriminating among all classes seen in all previous tasks without knowing task-IDs. Given task-IDs to the model during the test phase, CIL reduces to TIL (Ven & Tolias, 2019). The issue with CIL is that it not only suffers from catastrophic forgetting (CF) but also task confusion (TC), which happens when the model struggles to distinguish among different tasks observed so far while still being able to identify classes within a given task (Rebuffi et al., 2017; Belouadah et al., 2021; Masana et al., 2020; Cormerais et al., 2021).

Various strategies have been proposed for incremental learning such as regularization (Kirkpatrick et al., 2017; Zenke et al., 2017; Li & Hoiem, 2017b), bias-correction (Zeno et al., 2021), replay (Shin et al., 2017; Ven et al., 2020), and generative classifier

Table 1: Hybrid replay strategy versus baseline strategies.

| CIL Strategies | Memory | Compute | Performance |
|---|---|---|---|
| Generative Replay | $\mathcal{O}(cte)$ | $\mathcal{O}(t)$ | non-SOTA |
| Generative Classifier | $\mathcal{O}(t)$ | $\mathcal{O}(cte)$ | non-SOTA |
| Exemplar Replay | $\mathcal{O}(t)$ | $\mathcal{O}(t)$ | SOTA |
| **Hybrid Replay (ours)** | $\mathcal{O}(0.1t)$ | $\mathcal{O}(t)$ | **SOTA** |

(Ven et al., 2021; Pang et al., 2005; Zając et al., 2023). However, in the context of CIL, only the latter two strategies have been proven effective in overcoming TC: replay and generative classifier (Masana et al., 2020; Cormerais et al., 2021; Ven et al., 2021). Nevertheless, current realizations of the generative classifier strategy (Ven et al., 2021; Pang et al., 2005; Zając et al., 2023) demand an expanding architecture, leading to a linear increase of memory $\mathcal{O}(t)$ with respect to the number of learned tasks $t$. Furthermore, such expanding architectures fail to consolidate features of different tasks within a single model.

To develop a scalable incremental learning algorithm suitable for CIL, we capitalize on replay strategies (Shin et al., 2017; Ven et al., 2020). However, the current exemplar replay strategies (Rebuffi et al., 2017) are not scalable due to their reliance on large memory sizes. Specifically, the

memory requirements for exemplar replay increase linearly with the number of tasks, resulting in $\mathcal{O}(t)$ memory complexity (Hou et al., 2019; Wu et al., 2019; Hayes et al., 2020a).

To address this issue, generative replay strategies (Shin et al., 2017; Ven et al., 2020), instead of storing the exact data samples of the previous tasks, train a generative model to generate the pseudo-data pertaining to the previous tasks for replay to the discriminative model, achieving $\mathcal{O}(cte)$ memory complexity. Since the quality of the generated pseudo-data is not satisfactory, these strategies also undergo significant CF unless a very cumbersome generative model is trained which is inefficient. Consequently, generative replay diverts the problem of training a discriminative model incrementally to training a generative model incrementally which can be equally, if not more, challenging. (Ven et al., 2020).

The complexity of $\mathcal{O}(t)$ for either memory or compute is indispensable. *Because in principle every time an IL strategy learns task $t$, there have to be mechanisms at play to watch for $t-1$ conditions imposed by prior tasks on the weights of the neural networks lest those knowledge are overwritten. That requires either $\mathcal{O}(t)$ memory or $\mathcal{O}(t)$ compute complexity.* In Table 1, either case can be seen: while generative replay achieves $\mathcal{O}(cte)$ for memory, it nevertheless needs $\mathcal{O}(t)$ for computation. Conversely, the generative classifier is exactly the opposite: it achieves $\mathcal{O}(cte)$ for computation but requires $\mathcal{O}(t)$ for memory to accommodate the new tasks. Neither of these strategies is optimal. The performant strategy is exemplar replay which requires $\mathcal{O}(t)$ for both memory and compute.

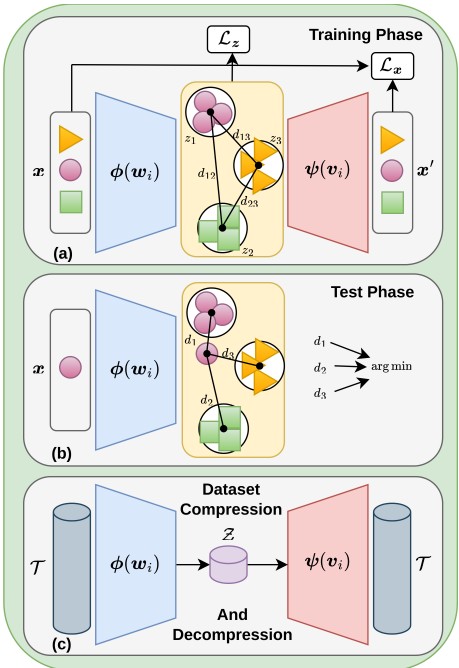

Figure 1: (a) Usage of RFA for the latent space. (b) Adoption of Euclidean distance during test. (c) HAE for compression and decompression of the dataset for replay.

We demonstrate that it is feasible to combine the strengths of both the exemplar (Rebuffi et al., 2017) and generative replay (Shin et al., 2017; Ven et al., 2020) in a *hybrid replay* strategy and consistently achieve state-of-the-art (SOTA) performance while significantly reducing the memory footprint of the exemplar replay strategy from $\mathcal{O}(t)$ to $\mathcal{O}(0.1t)$ at the worst case, using a new *hybrid autoencoder* based on *hybrid replay*. Our main contributions are as follows:

- We propose a novel autoencoder named hybrid autoencoder (HAE): the term hybrid autoencoder' indicates that HAE is capable of both discriminative and generative modeling (Goodfellow et al., 2016), for classification and replay, respectively. Furthermore, HAE employs the charged particle system energy minimization (CPSEM) equations and repulsive force algorithm (RFA) (Nazmitdinov et al., 2017) for the incremental embedding and distribution of new classes in its latent space. HAE uses RFA (Nazmitdinov et al., 2017) in its latent space to repel samples of different classes away from each other such that in the test phase the Euclidean distance can be used to measure the distance of a sample from centroids of different classes in the latent space to know to which class the test sample belongs to (see Figs. 1(a) and 1(b)).

- We propose a new strategy called autoencoder-based hybrid replay (AHR): the term hybrid replay' refers to the fact that AHR utilizes *a combination of exemplar replay and generative replay*. Specifically, AHR does not store the exemplars in the input space like exemplar replay which would require a large memory $\mathcal{O}(t)$; it rather stores the data samples in the latent space after they are encoded $\mathcal{O}(0.1t)$. Hence, AHR has characteristics that leverage the advantages of both exemplar and generative replay. AHR can decode data samples when they are needed for replay with negligible loss of fidelity because the decoder has been designed to *memorize* the training data as opposed to being designed to generalize and produce novel data samples. Fig. 1(c) shows how the data generation for replay is performed. As a result, AHR does not suffer from hazy *pseudo-data* during replay like

other generative replay strategies but has access to the *original* data. Table 1 contrasts the memory/compute complexities of ours and three baseline strategies. A detailed description of how we derived Table 1, along with relevant discussions, can be found in Appendix A.

- We provide comprehensive experiments to demonstrate the strong performance of AHR: we conduct our experiments across five benchmarks and ten baselines to showcase the effectiveness of AHR utilizing HAE and RFA while operating with the same memory and compute budgets.

We present our new strategy AHR in the following section. In Section 3, we contextualize AHR in the incremental learning literature. Section 4 details our evaluation methodology, including the baselines and benchmarks used, as well as the results of our experiments. Finally, Section 5 concludes this paper and provides future works for CIL.

## 2    OUR STRATEGY: AUTOENCODER-BASED HYBRID REPLAY (AHR)

In this section, we present our strategy AHR in a task-based CIL system model for simplicity and comparability with most of the works in the literature (Parisi et al., 2019; De Lange et al., 2021; Ven & Tolias, 2019; Masana et al., 2020; Zając et al., 2023). Nevertheless, our approach AHR is not restricted to the task-based CIL setting and can operate within the task-free setting as well (see Fig. 2) (Ven et al., 2021; Aljundi et al.,

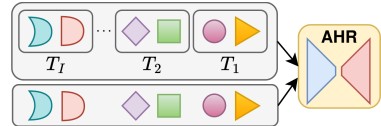

Figure 2: Task-based and task-free.

2019b). According to the task-based CIL system model, AHR visits a series of distinct non-repeating tasks $T_1, T_2, \ldots, T_I$. During each task $T_i$, a dataset $D_i := \{(\boldsymbol{x}_i^{j,k}, y_i^{j,k})\}_{j,k=1}^{J_i, K_i^j}$ is presented where $i$, $j$, and $k$ index task, class, and sample, bounded by $I$, $J_i$, and $K_i^j$. In the test phase, AHR must decide to which class a given input $\boldsymbol{x}_i^{j,k}$ belongs among all possible classes $\bigcup_{i=1}^I J_i$. Task-ID $i$ is not given during the test phase of CIL, indicating that AHR has to distinguish not only between classes that have been seen together in a given task but also between different tasks visited at different times.

**HAE.** AHR utilizes HAE consisting of an encoder and a decoder that can be formulated as follows: the encoder function $\boldsymbol{\phi}(\boldsymbol{x}_i^{j,k}) : \mathbb{R}^n \to \mathbb{R}^m$ maps the input data $\boldsymbol{x}_i^{j,k} \in \mathbb{R}^n$ to the *low-dimensional* latent representation $\boldsymbol{z}_i^{j,k} \in \mathbb{R}^m$. The decoder function $\boldsymbol{\psi}(\boldsymbol{z}_i^{j,k}) : \mathbb{R}^m \to \mathbb{R}^n$ reconstructs the input data from the latent representation. Note that AHR intentionally refuses to use the most popular autoencoders, Variational Autoencoders (VAE) (Kingma & Welling, 2013), since the goal here is not generalization or generating *new* images. Indeed, the goal of AHR is precisely the opposite: to have the decoder deterministically *memorize* pairs of $(\boldsymbol{z}_i^{j,k}, \boldsymbol{x}_i^{j,k})$. Compared to traditional autoencoders, HAE not only minimizes the reconstruction error between the input and the reconstructed data $\boldsymbol{L_x}(\boldsymbol{x}, \hat{\boldsymbol{x}})$ but also ensures that samples from the same class are clustered closely together in the latent space $\boldsymbol{L_z}(\boldsymbol{z}, \boldsymbol{p})$:

$$\boldsymbol{L}(\boldsymbol{x}, \hat{\boldsymbol{x}}, \boldsymbol{z}) = \boldsymbol{L_x}(\boldsymbol{x}, \hat{\boldsymbol{x}}) + \lambda \boldsymbol{L_z}(\boldsymbol{z}, \boldsymbol{p}) = \sum_{i=1}^I \sum_{j=1}^{J_i} \sum_{k=1}^{K_i^j} ||\boldsymbol{x}_i^{j,k} - \hat{\boldsymbol{x}}_i^{j,k}||^2 + \lambda ||\boldsymbol{z}_i^{j,k} - \boldsymbol{p}_i^j||^2 \quad (1)$$

where $|| \cdot ||^2$ denotes the $L^2$ norm, $\boldsymbol{p}_i^j$ is the $i$th task and $j$th class centroid embedding (CCE), and $\lambda$ is a hyperparameter. While the given loss function helps in clustering samples of the same class together, another crucial aspect is separating different classes. This separation, and in general, the incremental placement of CCEs in the latent space is addressed through CPSEM equations and RFA, where $\boldsymbol{p}_i^j$'s are akin to charged particles.

To model the energy dynamics within our system akin to charged particles, AHR employs the formulation based on Coulomb interaction energy. Consider a fixed set of $I \times \sum_i J_i$ particles representing CCEs, with charges $q_i^j$ at positions $\boldsymbol{p}_i^j$. The potential energy of this system is given by

$$\mathcal{U} = \sum_{i,j=1}^{I,J_i} \frac{(q_i^j)^2}{2} \sum_{i',j' \neq i,j} \frac{1}{\|\boldsymbol{p}_{i'}^{j'} - \boldsymbol{p}_i^j\|}. \quad (2)$$

Each particle also possesses kinetic energy $\mathcal{K}_i^j = \frac{1}{2} m_i^j \|\boldsymbol{v}_i^j\|^2$, where $m_i^j$ and $\boldsymbol{v}_i^j$ represent the mass and speed of particle $ij$ (CCE of task $i$ and class $j$), respectively. In our optimization framework,

AHR aims to minimize the total energy $\mathcal{E} = \mathcal{U} + \mathcal{K}$, where $\mathcal{K} = \sum_{k=i,j}^{I,J_i} \frac{1}{2} m_i^j \|\boldsymbol{v}_i^j\|^2$. This can be achieved through the calculus of variations, leveraging the Lagrangian:

$$\mathcal{L} = \mathcal{K} - \mathcal{U} = \sum_{k=i,j}^{I,J_i} \frac{1}{2} m_i^j \|\boldsymbol{v}_i^j\|^2 - \sum_{k=i,j}^{I,J_i} \frac{(q_i^j)^2}{2} \sum_{i',j' \neq i,j} \frac{1}{\|\boldsymbol{p}_{i'}^{j'} - \boldsymbol{p}_i^j\|}. \tag{3}$$

The equations of motion for the particles, derived via the Euler-Lagrange equation

$$\frac{d}{dt}\left(\frac{\partial \mathcal{L}}{\partial \boldsymbol{v}_i^j}\right) = \frac{\partial \mathcal{L}}{\partial \boldsymbol{p}_i^j} \tag{4}$$

enable us to determine the optimal positions of the particles, effectively minimizing the total energy of our system. The above equation helps HAE to efficiently distribute CCEs within the latent space.

**AHR.** Algorithm 1 outlines the steps of the AHR strategy in the context of task-based CIL: Whenever a new task $T_\ell$ arrives AHR invokes three main routines of CCE_PLACEMENT, HAE_TRAIN, and MEMORY_POPULATION: (i) in CCE_PLACEMENT, AHR determines the positions of the new CCEs for $D_\ell$ and returns $\mathcal{P}_\ell = \{\boldsymbol{p}_\ell^j\}_{j=1}^{J_\ell}$ based on RFA outlined in Algorithm 2 solving Eq. 4 where $\ell$ denotes the latest arrived task (throughout this paper, we use the notation $\ell$ referring to the latest task index whereas $i$ might refer to any task). Note the distinction that, unlike the centroids in iCaRL (Rebuffi et al., 2017), CCEs in AHR do not change over the course of learning.

(ii) After the placement of the CCEs $\mathcal{P}_\ell = \{\boldsymbol{p}_\ell^j\}_{j=1}^{J_\ell}$, HAE_TRAIN is invoked by AHR (outlined in Algorithm 3) where the model $\{\boldsymbol{\phi}(\boldsymbol{w}_{\ell-1}), \boldsymbol{\psi}(\boldsymbol{v}_{\ell-1})\}$ is copied as $\{\boldsymbol{\phi}(\boldsymbol{w}_\ell), \boldsymbol{\psi}(\boldsymbol{v}_\ell)\}$. While $\{\boldsymbol{\phi}(\boldsymbol{w}_{\ell-1}), \boldsymbol{\psi}(\boldsymbol{v}_{\ell-1})\}$ is kept frozen, $\{\boldsymbol{\phi}(\boldsymbol{w}_\ell), \boldsymbol{\psi}(\boldsymbol{v}_\ell)\}$ is trained on the combined training set featuring the new tasks and the decoded exemplars $D \leftarrow D_\ell \cup \boldsymbol{\psi}(\boldsymbol{v}_{\ell-1}, \{\mathcal{M}_i\}_{i=1}^{\ell-1})$ for $E$ number of epochs with the loss function in Eq. 1 and a distillation loss (serving as a data regularization strategy) to obtain $\{\boldsymbol{\phi}(\boldsymbol{w}_\ell), \boldsymbol{\psi}(\boldsymbol{v}_\ell)\}$, where the memory $\mathcal{M}_{\ell-1} = \{\boldsymbol{e}_{\ell-1}^{j,k}\}_{j,k=1}^{J_{\ell-1}, K_{\ell-1}^j}$ stores the

---

**Algorithm 1: AUTOENCODER-BASED HYBRID REPLAY**

**Input:** Tasks $\{T_1, T_2, \ldots, T_I\}$ with $\{D_1, D_2, \ldots, D_I\}$, HAE model $\{\boldsymbol{\phi}(\boldsymbol{w}_0), \boldsymbol{\psi}(\boldsymbol{v}_0)\}$, memory $\mathcal{M}_0$
**Output:** $\{\boldsymbol{\phi}(\boldsymbol{w}_I^*), \boldsymbol{\psi}(\boldsymbol{v}_I^*)\}$, $\{\mathcal{M}_i^*\}_{i=1}^I$, CCEs $\{\mathcal{P}_i^*\}_{i=1}^I$
**for** tasks $i = 1, \ldots, I$ **do:**
  $\mathcal{P}_i = \text{CCE\_PLACEMENT}(\boldsymbol{\phi}(\boldsymbol{w}_{i-1}), \boldsymbol{\psi}(\boldsymbol{v}_{i-1}), D_i,$
    $\{\mathcal{P}_{i'}\}_{i'=1}^{i-1})$ via RFA in Algorithm 2 solving Eq. 4
  $\boldsymbol{\phi}(\boldsymbol{w}_i), \boldsymbol{\psi}(\boldsymbol{v}_i) = \text{HAE\_TRAIN}(\boldsymbol{\phi}(\boldsymbol{w}_{i-1}), \boldsymbol{\psi}(\boldsymbol{v}_{i-1}),$
    $D_i, \{\mathcal{M}_{i'}\}_{i'=1}^{i-1}, \{\mathcal{P}_{i'}\}_{i'=1}^i)$ via Algorithm 3
  $\mathcal{M}_i = \text{MEMORY\_POPULATION}(\boldsymbol{\phi}(\boldsymbol{w}_i), \boldsymbol{\psi}(\boldsymbol{v}_i), \boldsymbol{\phi}(\boldsymbol{w}_{i-1}),$
    $\boldsymbol{\psi}(\boldsymbol{v}_{i-1}), D_i, \{\mathcal{M}_{i'}\}_{i'=1}^{i-1}, \{\mathcal{P}_{i'}\}_{i'=1}^i))$ via Algorithm 4
  Delete $\boldsymbol{\phi}(\boldsymbol{w}_{i-1}), \boldsymbol{\psi}(\boldsymbol{v}_{i-1})$
**end for**

---

**Algorithm 2: CCE_PLACEMENT**

**Input:** $\{\mathcal{P}_1, \mathcal{P}_2, \ldots, \mathcal{P}_{\ell-1}\}$, Repulsive Constant $\zeta$, Particle Mass $m$, Time Step $\Delta t$, Simulation Duration $\tau$
**Output:** $\mathcal{P}_\ell$
$\mathcal{P}_\ell^{t=0} = \boldsymbol{\phi}(\boldsymbol{w}_{\ell-1}, D_i)$ // initialize $\mathcal{P}_\ell$
**for** time step $t = 1, \ldots, \tau$ **do:**
  **for** classes $j = 1, \ldots, J_\ell$ **do:**
    **for** tasks $i = 1, \ldots, \ell$ **do:**
      **for** classes $j' = 1, \ldots, J_\ell$ **do:**
        **if** $i, j' \neq \ell, j$ **then:**
          Compute displacement vector $\boldsymbol{d}_{\ell i'}^{jj'} = \boldsymbol{p}_\ell^j - \boldsymbol{p}_{i'}^{j'}$
          Compute repulsive force $\boldsymbol{f}_{\ell i'}^{jj'} = \frac{\zeta}{|\boldsymbol{d}_{\ell i'}^{jj'}|^2} \cdot \frac{\boldsymbol{d}_{\ell i'}^{jj'}}{|\boldsymbol{d}_{\ell i'}^{jj'}|}$
          Accumulate repulsive force: $\boldsymbol{F}_\ell^j = \boldsymbol{F}_\ell^j + \boldsymbol{f}_{\ell i'}^{jj'}$
      **end for**
    **end for**
    Update CCE velocity: $\boldsymbol{v}_\ell^j = \boldsymbol{v}_\ell^j + \frac{\boldsymbol{F}_\ell^j}{m} \cdot \Delta t$
    Update CCE position: $\boldsymbol{p}_\ell^j = \boldsymbol{p}_\ell^j + \boldsymbol{v}_\ell^j \cdot \Delta t$
  **end for**
**end for**

---

**Algorithm 3: HAE_TRAIN**

**Input:** $\boldsymbol{\phi}(\boldsymbol{w}_{\ell-1}), \boldsymbol{\psi}(\boldsymbol{v}_{\ell-1}), D_\ell, \{\mathcal{P}_i\}_{i=1}^\ell, \{\mathcal{M}_i\}_{i=1}^{\ell-1}$
**Output:** $\boldsymbol{\phi}(\boldsymbol{w}_\ell^*), \boldsymbol{\psi}(\boldsymbol{v}_\ell^*)$
$D \leftarrow D_\ell \cup \boldsymbol{\psi}(\boldsymbol{v}_{\ell-1}, \mathcal{M}_{\ell-1})$
Copy $\boldsymbol{\phi}(\boldsymbol{w}_{\ell-1}), \boldsymbol{\psi}(\boldsymbol{v}_{\ell-1})$ as $\boldsymbol{\phi}(\boldsymbol{w}_\ell), \boldsymbol{\psi}(\boldsymbol{v}_\ell)$
**for** epochs $e = 1, \ldots, E$ **do:**
  **for** minibatch $b = 1, \ldots, B$ **do:**
    minimize the HAE losses in Eq. 1 and Distillation losses

$$\boldsymbol{L}(D, \boldsymbol{\psi}(\boldsymbol{v}_\ell, \boldsymbol{\phi}(\boldsymbol{w}_\ell, D)), \boldsymbol{\phi}(\boldsymbol{w}_\ell, D)) +$$
$$\|\boldsymbol{\phi}(\boldsymbol{w}_{\ell-1}, D) - \boldsymbol{\phi}(\boldsymbol{w}_\ell, D)\| +$$
$$\|\boldsymbol{\psi}(\boldsymbol{v}_{\ell-1}, \boldsymbol{\phi}(\boldsymbol{w}_{\ell-1}, D)) - \boldsymbol{\psi}(\boldsymbol{v}_\ell, \boldsymbol{\phi}(\boldsymbol{w}_\ell, D))\|$$

    with an arbitrary optimizer to obtain $\boldsymbol{\phi}(\boldsymbol{w}_\ell^*), \boldsymbol{\psi}(\boldsymbol{v}_\ell^*)$
  **end for**
**end for**

---

exemplars of task $\ell - 1$ and $\boldsymbol{e}_{\ell-1}^{j,k}$ contains exemplar $(\ell-1)jk$ coupled with its label. In AHR, the memory $\mathcal{M}_{\ell-1}$ stores embedded vectors in the latent space, not raw data samples. In practice,

for each iteration of the SGD, besides $1/\ell$ fraction of the minibatch size that is provided by the new task $T_\ell$, $(\ell - 1)/\ell$ fraction of minibatch size is selected and instantly decoded from memory $\psi(v_{\ell-1}, \{\mathcal{M}_i\}_{i=1}^{\ell-1})$ in a statistically representative fashion with respect to the previous tasks/classes for training. These vectors require significantly less memory $\mathcal{O}(0.1 \times t)$ than raw data and can be decoded at any time by $\psi(v_{\ell-1}, \{\mathcal{M}_i\}_{i=1}^{\ell-1})$.

(iii) AHR populates its memory $\mathcal{M}_\ell$ via MEMORY_POPULATION in Algorithm 4 based on Herding as in (Rebuffi et al., 2017). Currently, there are two competitive approaches for sampling of exemplars in the literature (Masana et al., 2020), Herding and Naive Random, where the Herding approach on average demonstrates a slight improvement over Naive Random sampling when applied to longer sequences of tasks (Masana et al., 2020). AHR, in Algorithm 4, computes the losses $L_z(\phi(w_\ell, D), \{\mathcal{P}_i\}_{i=1}^\ell)$ and ranks

---
**Algorithm 4:** MEMORY_POPULATION

**Input:** $\phi(w_\ell)$, $\psi(v_\ell)$, $\phi(w_{\ell-1})$, $\psi(v_{\ell-1})$, $D_\ell$, $\{\mathcal{P}_i\}_{i=1}^\ell$, $\{\mathcal{M}_i\}_{i=1}^{\ell-1}$

**Output:** $\{\mathcal{M}_i\}_{i=1}^\ell$

$\varepsilon = M/(\ell \times \sum_{j=1}^\ell J_j)$ // memory size / # of classes so far

$D \leftarrow D_\ell \cup \psi(v_{\ell-1}, \{\mathcal{M}_i\}_{i=1}^{\ell-1})$

Calculate the losses $L_z(\phi(w_\ell, D), \{\mathcal{P}_i\}_{i=1}^\ell)$ as in Eq. 1

**for** tasks $i = 1, \ldots, \ell$ **do:**
    **for** classes $j = 1, \ldots, J_\ell$ **do:**
        **for** samples $k = 1, \ldots, \varepsilon$ **do:**
            Store RANK$(L_z(\phi(w_i, D), \{\mathcal{P}_i\}_{i=1}^\ell))_i^{j,k}$ as $e_i^{j,k}$
        **end for**
    **end for**
**end for**

---

them ascendingly via RANK and then selects $\varepsilon$ number of classes for both the new task $\ell$ and previous ones.

Finally, at the test stage, it is determined to which task-class $ij$ a given data sample $x$ belongs via computing the Euclidean distance as follows: $\arg\min_{i,j} \|\phi(w_I^*, x) - p_i^j\|$. It is clear that in CIL, not only must the classes within a given task be discriminated, but the different tasks themselves must also be distinguished, which is why TC takes place on top of CF.

## 3 LITERATURE REVIEW AND CONTEXTUALIZATION OF AHR

**Task-based or task-free.** Incremental learning literature features *various learning scenarios* that present their own unique challenges, and accordingly, *diverse strategies* have been developed (Parisi et al., 2019; De Lange et al., 2021). In the first place, there are two learning scenarios of task-based (Ven & Tolias, 2019; Masana et al., 2020) and task-free (Ven et al., 2021; Aljundi et al., 2019b). In task-based, the model receives the data in the form of tasks: The task-based scenario is divided into two popular scenarios: task-incremental learning (TIL) and class-incremental learning (CIL). Whereas in TIL the model receives the task-IDs during both training and inference, in CIL the task-IDs are not given during inference and the model must infer them. The task-free scenario, meanwhile, totally abandons the notion of tasks altogether (Ven et al., 2021; Aljundi et al., 2019b). AHR makes no prohibitive assumptions and can operate in all the above scenarios. Nevertheless, for comparability with the majority of works in IL, *this paper examines the performance of AHR in the CIL setting*.

**Offline or online.** Incremental learning can be either offline (Masana et al., 2020) or online (Zając et al., 2023), where in the offline learning scenario, the data of each task can be fed to the model multiple times before moving on to the next task. Conversely, in the online scenario, the model visits the data only once as they arrive and cannot iterate on them. In the literature, there are more works on the offline scenario (Parisi et al., 2019; De Lange et al., 2021; Masana et al., 2020). Therefore, *this paper examines the performance of AHR in the offline scenario*.

**Challenges and strategies.** CIL faces many challenges such as CF, weight drift, activation drift, task-recency bias, and TC (Masana et al., 2020; Cormerais et al., 2021). To tackle the weight drift and activation drift challenges of CIL (Kao et al., 2020), the most popular category of strategies, regularization, is often adopted (Zenke et al., 2017). Although regularization is not alone effective in mitigating the TC challenge of CIL (Ven et al., 2021), it has been proven productive in tandem with other strategies like exemplar strategies (Rebuffi et al., 2017; Farquhar & Gal, 2018a; Hsu et al., 2018; Castro et al., 2018; Dhar et al., 2019; Serra et al., 2018). Regularization strategies have two branches:

**Weight regularization and data regularization.** Weight regularization mitigates weight drift of the parameters optimized for the previous tasks by assigning an importance coefficient for each parameter in the network (assuming the independence of weights) after learning each task (Kirkpatrick et al., 2017; Zenke et al., 2017; Nguyen et al., 2018; Farquhar & Gal, 2018b; Aljundi et al., 2018; Chaudhry et al., 2018). When learning new tasks, the importance coefficients help in minimizing weight drift. EWC (Kirkpatrick et al., 2017) and SI (Zenke et al., 2017) are popular weight regularization strategies. EWC (Kirkpatrick et al., 2017) relies on a diagonal approximation of the Fisher matrix to weigh the contributions from different parameters. SI (Zenke et al., 2017) maintains and updates per-parameter importance measures in an online manner.

Data regularization is the second regularization strategy, aimed at preventing activation drift through knowledge distillation (Buciluundefined et al., 2006; Hinton et al., 2015), originally designed to learn a more parameter-efficient student network from a larger teacher network. Differs from weight regularization which imposes constraints on parameter updates, knowledge distillation focuses on ensuring consistency in the responses of the new and old models. This distinctive feature provides a broader solution space which is why distillation has become the dominant strategy used in tandem with rehearsal strategy (Li & Hoiem, 2017a; Jung et al., 2016; Dhar et al., 2019; Zhang et al., 2020; Lee et al., 2019). LwF (Li & Hoiem, 2017a), a popular data regularization strategy, uses a distillation loss to keep predictions consistent with the ones from an old model. *Our AHR strategy as discussed in the previous section incorporates LwF into exemplar replay to overcome activation drift.*

**Bias-correction.** Its aim is to address the challenge of task-recency bias, which refers to the tendency of incrementally learned networks to be biased towards classes in the most recently learned task (Belouadah et al., 2021; Li et al., 2020; Wu et al., 2019; Maltoni & Lomonaco, 2019; Lomonaco & Maltoni, 2017; Zeno et al., 2021; Belouadah & Popescu, 2020). Labels trick (LT) (Dhar et al., 2019) is a rehearsal-free bias-correcting algorithm that prevents negative bias on past tasks. LT often improves the performance when added on top of other strategies (Wu et al., 2019). However, *because AHR separates representation learning from classification similar to* (Rebuffi et al., 2017), *which gives a dose of immunity to task-recency bias. Because AHR samples the tasks/classes for its minibatch in a statistically representative manner during training inspired by* (Castro et al., 2018), *incorporating an explicit bias-correction strategy such as LT proved unnecessary.*

**Generative classifier.** CIL strategies can be categorized into two types: discriminative- and generative-based classification. Strategies count as discriminative classifiers when eventually a discriminator performs the classification. However, generative classifiers perform classification only and directly using generative modeling (Ven et al., 2021; Pang et al., 2005; Zając et al., 2023). Gen-C (Ven et al., 2021) is a novel rehearsal-free generative classifier strategy; the strategy does energy-based generative modeling (Li et al., 2020) via VAEs (Kingma & Welling, 2013) and importance sampling (Burda et al., 2016). SLDA (Hayes & Kanan, 2020) (popular in data mining (Kim et al., 2011; Pang et al., 2005)) is thought to be another form of generative classifier (Ven et al., 2021); however, it prevents representation learning.

**Exemplar replay.** The most popular exemplar replay strategy, iCaRL (Rebuffi et al., 2017), fuses exemplar replay and LwF (Li & Hoiem, 2017a). To mitigate the task-recency bias, iCaRL *separates* the classifier from the representation learning (implicit bias-correction). At inference, iCaRL classifies via the closest centroid. EEIL (Castro et al., 2018) introduces *balanced training*, where at the end of each training session an equal number of exemplars from all classes is used for a limited number of iterations. Similar to iCaRL, EEIL incorporates data regularization (distillation loss) into exemplar replay. *As discussed in the previous section, our strategy, AHR, also separates the representation learning and classification. Furthermore, AHR employs balanced training (from EEIL) and distillation loss via LwF (similar to iCaRL).*

MIR (Aljundi et al., 2019a) trains on the exemplar memory by selecting exemplars that have a larger loss increase after each training step. GD (Lee et al., 2019) utilizes external data to distill knowledge from previous tasks. BiC (Wu et al., 2019) learns to rectify the bias in predictions by adding an additional layer while IL2M (Belouadah & Popescu, 2019) introduces saved certainty statistics for predictions of classes from previous tasks (explicit bias-correction). LUCIR (Hou et al., 2019) replaces the standard softmax layer with a cosine normalization layer. GDumb (Prabhu et al., 2020) ensures a balanced training set. ER (Chaudhry et al., 2019) analyzes the effectiveness of episodic memory.

**Generative replay (pseudo-rehearsal).** This strategy generates synthetic examples of previous tasks via a generative model trained on previous tasks. DGR (Shin et al., 2017) generates synthetic samples via an unconditional GAN where an auxiliary classifier classifies synthetic samples. MeRGAN (Wu et al., 2018) improves DGR via a label-conditional GAN and replay alignment. DGM (Ostapenko et al., 2019) combines the advantages of conditional GANs and synaptic plasticity using neural masking leveraging dynamic network expansion mechanism to increase model capacity. Lifelong GAN (Zhai et al., 2019) extends image generation without CF from label-conditional to image-conditional GANs. Other strategies perform feature replay (Ven et al., 2020; Xiang et al., 2019; Kemker & Kanan, 2017) which needs a fixed backbone network to provide good representations.

**Hybrid replay (AHR).** Technically, our AHR strategy lies somewhere between *exemplar replay* and *generative replay*: AHR differs from exemplar replay in that it stores the samples in the latent space. AHR also differs from generative replay in that it does not rely on synthetic data or pseudo-data; instead, it relies on the memorization of the original data. By encoding and decoding the exemplars into and out of the latent space, AHR mitigates the memory complexity of current exemplar replay strategies significantly and can be readily applied to the exemplar replay strategies. In the literature, hybrid replay has been studied in several works (Zhou et al., 2022): Tong et al. (2022) utilizes a closed-loop encoding-decoding framework that stores only the means and covariances of features rather than the individual features themselves. The authors organize the latent space using a Linear Discriminative Representation, which partitions the latent space into a series of linear subspaces, each corresponding to a distinct class of objects. Hayes et al. (2020b); Wang et al. (2021) do not store raw exemplars but instead using compressed exemplars derived from mid-level CNN features (a strategy that is increasingly recognized as a promising direction in incremental learning research). In (Hayes et al., 2020b; Wang et al., 2021), classification occurs after the decoding process, employing a cross-entropy loss function. In contrast, our method performs classification directly within the latent space of the encoder, similar to (Tong et al., 2022). Pellegrini et al. (2020) stores 'activations volumes' of intermediate layers rather than raw data.

# 4 EXPERIMENTAL RESULTS

**Baselines.** We consider five categories of baselines: vanilla strategies, generative classifier, generative replay, exemplar replay, and hybrid replay. Vanilla strategies include the naive strategies that serve as the lower or upper bound: Fine-Tuning (FT) does simple fine-tuning whenever a new task arrives, FT-E incorporates exemplar replay into fine-tuning, and Joint trains on all tasks seen so far (Masana et al., 2020). For generative classifiers, we include SLDA (Hayes & Kanan, 2020), Gen-C (Ven et al., 2021), and PEC (Zając et al., 2023). For generative replay, DGR (Shin et al., 2017), MeRGAN (Wu et al., 2018), and BI-R-SI (Ven et al., 2020) are considered. For exemplar replay, the most strong baseline strategy, we include GD (Lee et al., 2019), GDumb (Prabhu et al., 2020), iCaRL (Rebuffi et al., 2017), EEIL (Castro et al., 2018), BiC (Wu et al., 2019), LUCIR (Hou et al., 2019), and IL2M (Belouadah & Popescu, 2019). Finally, for hybrid replay, we include i-CTRL (Tong et al., 2022), REMIND (Hayes et al., 2020b), and REMIND+ (Wang et al., 2021). Note that since regularization and bias-correction strategies have been often applied to the aforementioned strategies supplementarily (and they are not performant for CIL on their own), we do not provide results for them independently.

**Benchmarks.** The series of tasks for CIL are constructed according to (Masana et al., 2020; Ven et al., 2021; Zając et al., 2023), where the popular image classification datasets are split up such that each task presents data pertaining to a subset of classes, in a non-overlapping manner. For naming benchmarks, we follow (Masana et al., 2020), where dataset $D$ is divided into $T$ tasks with $C$ classes for each task. Hence, a benchmark is named as $D(T/C)$. Accordingly, we have *MNIST*$(5/2)$ (LeCun et al., 2010), *Balanced SVHN*$(5/2)$ (Netzer et al., 2011), *CIFAR-10*$(5/2)$ (Krizhevsky et al., 2009), *CIFAR-100*$(10/10)$ (Krizhevsky et al., 2009), and *miniImageNet*$(20/5)$ (Vinyals et al., 2016) benchmarks. **Metrics.** Performance is evaluated by the final test accuracy after training on the series of all tasks.

**Network architectures.** For MNIST benchmark, as in (Ven et al., 2021), a dense network with 2 hidden layers of 400 ReLU units was used. We utilized its mirror for the decoder. For larger datasets, as suggested by Masana et al. (2020); He et al. (2016), ResNet-32 is used. We employed 3 layers of CNNs for the decoder. **Hyperparameters.** We adopt Adam (Kingma & Ba, 2014) as the optimizer.

Table 2: Empirical evaluation of AHR against a suite of CIL baselines. Accuracies and the SEMs.

| Benchmarks Tasks Conf. #Total Exemplars | BC | Reg | MNIST (5/2) 200 | BalancedSVHN (5/2) 200 | CIFAR-10 (5/2) 200 | CIFAR-100 (10/10) 2000 | miniImageNet (20/5) 2000 |
|---|---|---|---|---|---|---|---|
| *Vanilla Strategies (Lower and Upper Bounds)* | | | | | | | |
| FT | N | N | $19.93_{\pm0.03}$ | $19.19_{\pm0.04}$ | $18.72_{\pm0.30}$ | $8.91_{\pm0.12}$ | $4.32_{\pm0.06}$ |
| FT-E | I | N | $92.17_{\pm0.16}$ | $87.13_{\pm0.37}$ | $72.17_{\pm0.84}$ | $48.47_{\pm0.83}$ | $39.02_{\pm0.74}$ |
| Joint | N | N | $98.48_{\pm0.06}$ | $95.88_{\pm0.04}$ | $92.37_{\pm0.09}$ | $73.87_{\pm0.10}$ | $73.45_{\pm0.15}$ |
| *Generative Classifier Strategies (Dynamically Expanding Architectures)* | | | | | | | |
| SLDA | N | N | $87.30_{\pm0.02}$ | $42.32_{\pm0.04}$ | $38.34_{\pm0.04}$ | $25.83_{\pm0.01}$ | $19.03_{\pm0.02}$ |
| Gen-C | N | N | $89.19_{\pm0.05}$ | $51.92_{\pm0.59}$ | $49.38_{\pm0.37}$ | $29.69_{\pm0.62}$ | $22.57_{\pm0.40}$ |
| PEC | N | N | $90.81_{\pm0.06}$ | $55.61_{\pm0.21}$ | $52.41_{\pm0.33}$ | $37.53_{\pm0.41}$ | $28.39_{\pm0.36}$ |
| *Generative Replay Strategies (Rehearsal of Pseudo-Exemplars)* | | | | | | | |
| DGR | I | N | $88.50_{\pm0.43}$ | $28.17_{\pm1.27}$ | $25.43_{\pm1.72}$ | $9.20_{\pm1.25}$ | $6.59_{\pm1.13}$ |
| MeRGAN | I | N | $89.83_{\pm0.37}$ | $33.49_{\pm1.35}$ | $27.17_{\pm1.84}$ | $11.39_{\pm1.23}$ | $7.82_{\pm1.05}$ |
| BI-R-SI | I | W | - | $38.32_{\pm1.43}$ | $37.48_{\pm1.96}$ | $34.37_{\pm1.20}$ | $29.71_{\pm1.03}$ |
| *Exemplar Replay Strategies (Rehearsal of Exemplars)* | | | | | | | |
| GD | I | D | $92.02_{\pm0.17}$ | $89.11_{\pm0.56}$ | $71.13_{\pm0.72}$ | $49.01_{\pm0.86}$ | $38.47_{\pm0.62}$ |
| GDumb | I | D | $91.13_{\pm0.19}$ | $88.02_{\pm0.47}$ | $72.79_{\pm0.50}$ | $47.29_{\pm0.71}$ | $39.64_{\pm0.70}$ |
| iCaRL | I | D | $93.06_{\pm0.33}$ | $89.63_{\pm0.61}$ | $73.29_{\pm0.73}$ | $49.38_{\pm0.62}$ | $43.51_{\pm0.68}$ |
| EEIL | E | D | $93.88_{\pm0.39}$ | $90.75_{\pm0.53}$ | $73.85_{\pm0.84}$ | $51.03_{\pm0.75}$ | $41.09_{\pm0.54}$ |
| BiC | E | D | $94.13_{\pm0.25}$ | $91.04_{\pm0.63}$ | $75.01_{\pm0.93}$ | $51.41_{\pm0.88}$ | $44.80_{\pm0.57}$ |
| LUCIR | I | D | $92.62_{\pm0.29}$ | $87.01_{\pm0.44}$ | $71.52_{\pm0.71}$ | $47.08_{\pm0.94}$ | $36.95_{\pm0.79}$ |
| IL2M | E | W | $94.07_{\pm0.21}$ | $90.64_{\pm0.57}$ | $73.86_{\pm0.78}$ | $50.06_{\pm0.73}$ | $43.64_{\pm0.59}$ |
| *Hybrid Exemplar-Generative Strategy (Rehearsal of Decoded Exemplars)* | | | | | | | |
| i-CTRL | I | D | $94.31_{\pm0.27}$ | $91.07_{\pm0.40}$ | $74.61_{\pm0.61}$ | $51.74_{\pm0.69}$ | $44.78_{\pm0.68}$ |
| REMIND | I | D | $93.95_{\pm0.19}$ | $91.38_{\pm0.64}$ | $75.02_{\pm0.65}$ | $50.93_{\pm0.75}$ | $43.92_{\pm0.71}$ |
| REMIND+ | I | D | $95.62_{\pm0.33}$ | $92.15_{\pm0.72}$ | $75.49_{\pm0.70}$ | $52.36_{\pm0.77}$ | $45.02_{\pm0.65}$ |
| AHR | I | D | $\mathbf{97.53}_{\pm0.32}$ | $\mathbf{93.02}_{\pm0.65}$ | $\mathbf{77.12}_{\pm0.75}$ | $\mathbf{54.43}_{\pm0.93}$ | $\mathbf{48.09}_{\pm0.64}$ |
| *AHR Ablation Study (Impact of Compression and RFA)* | | | | | | | |
| AHR-lossy-mini | | | $93.35_{\pm0.32}$ | $90.40_{\pm0.58}$ | $73.28_{\pm0.47}$ | $50.29_{\pm0.90}$ | $42.39_{\pm0.64}$ |
| AHR-lossless-mini | | | $93.76_{\pm0.26}$ | $90.88_{\pm0.50}$ | $73.68_{\pm0.41}$ | $50.85_{\pm0.81}$ | $42.88_{\pm0.59}$ |
| AHR-lossless | | | $98.12_{\pm0.08}$ | $94.21_{\pm0.23}$ | $78.35_{\pm0.37}$ | $56.71_{\pm0.57}$ | $49.70_{\pm0.32}$ |
| AHR-contrastive | | | $95.12_{\pm0.29}$ | $91.43_{\pm0.54}$ | $74.87_{\pm0.47}$ | $51.98_{\pm0.76}$ | $44.60_{\pm0.53}$ |
| AHR-GMM | | | $92.49_{\pm0.23}$ | $88.70_{\pm0.50}$ | $72.63_{\pm0.39}$ | $49.48_{\pm0.71}$ | $42.52_{\pm0.48}$ |

**Exemplar rehearsal.** All the strategies *always* follow the *fixed exemplar memory*, not *growing exemplar memory*, implying that the number of exemplars per class decreases over time to keep the overall memory size constant (Masana et al., 2020; Rebuffi et al., 2017). The learning rates, batch sizes, and strategy-dependent hyperparameters are detailed in Appendix B.

Table 2 demonstrates that hybrid approaches on average outperform alternative baselines on all five benchmarks (for a fixed compute, matched parameter count, and equal memory size for exemplar replay to ensure fairness in comparisons). Although hybrid approaches are given the same memory size, they can store more exemplars, depending on the compression rate given in Table 2 for each benchmark. The performance improvement of hybrid approaches, compared to exemplar rehearsal-based strategies, can be attributed to exemplar diversity (availability of more exemplars). The effectiveness of the availability of more exemplars, exemplar diversity, is well-documented in the literature (Masana et al., 2020). Notably, our AHR approach outperforms other hybrid replay methods. This superior performance is due to AHR's more effective embedding in the latent space using RFA, in contrast to i-CTRL, which relies on Linear Discriminative Representation. Additionally, AHR's architecture offers an advantage by classifying directly in the latent space, whereas REMIND and REMIND+ perform classification only after the decoding process.

**Ablating the impact of lossy compression.** As shown in Table 2, in the AHR-lossy-mini and AHR-lossless-mini settings, AHR is given as many exemplars as other exemplar rehearsal-based baselines, with imperfect and perfect quality. In the AHR-lossless setting, AHR is given as many exemplars as AHR, but with perfect quality. The aim of these three settings is to investigate how much *compression by the encoder*, and therefore *the opportunity to store more examples* benefit AHR. In the AHR-lossy-mini and AHR-lossless-mini settings, AHR performs on par with other exemplar

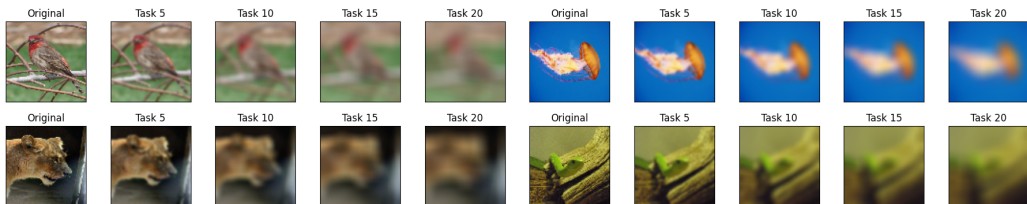

Figure 3: Images produced by the decoder at different tasks (for the decoder size of 1.8M).

rehearsal-based strategies. Interestingly, the performance loss in these settings is significantly greater than the performance gain in the AHR-lossless setting. This observation supports two key findings: (a) more exemplars, even decoded exemplars, significantly enhance performance (Masana et al., 2020); and (b) for mitigating CF, decoded exemplars are nearly as effective as perfect exemplars (Ven et al., 2020). Fig. 3 shows both the original and decoded images at different tasks (miniImageNet).

**Ablating the approaches for structuring the latent space.** Hybrid approaches such as REMIND and REMIND+ do not impose any explicit structure on the latent space. As a result, our discussion here will not cover these methods, focusing instead on techniques that structure the latent space. Among these, i-CTRL employs Linear Discriminative Representation, whereas AHR utilizes RFA. The comparative analysis presented in the latter rows of Table 2 also considers alternative structuring methods, such as contrastive loss (Cha et al., 2021) and the Gaussian Mixture Model (Ven et al., 2020). Our findings, as detailed in Table 2, indicate that RFA outperforms these alternatives. This is because RFA systematically embeds the class centroids of new classes into the latent space with minimal amount of shift from their natural position and minimum changes to the weights of the neural network achieved by CPSEM. This level of systematic embedding, necessary for ensuring the structuredness of the latent space, is not possible in alternative approaches.

**Bias-correction and regularization.** Table 2 also outlines the bias-correction and regularization methods used by different strategies. For bias-correction, "N," "I," and "E" represent none, implicit, and explicit, respectively. Implicit bias-correction, as seen in (Castro et al., 2018), relies on data equalization without directly manipulating the weights, whereas explicit correction, as in (Belouadah & Popescu, 2019), involves direct weight adjustment. For regularization, "N," "D," and "W" denote none, data regularization, and weight regularization, respectively. Most exemplar rehearsal-based baselines, along with our AHR strategy, use implicit bias-correction via data equalization and data distillation for regularization.

**Resource-consumption.** Fig. 5 assesses the performances for three benchmarks: *CIFAR-10*(5/2), *CIFAR-100*(10/10), and *miniImageNet*(20/5). It explores the relationships between (i) exemplar memory size and performance, (ii) exemplar memory size and compute time (epochs) for a target performance, and (iii) performance for a range of allocated compute times

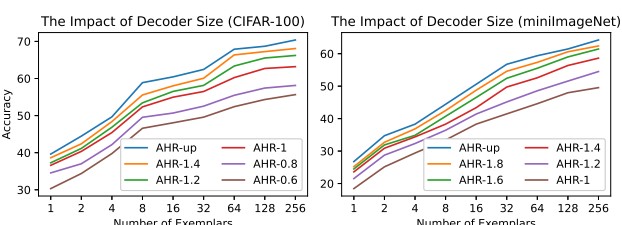

Figure 4: Performances for various decoder/memory sizes.

(wall-clock time), presented in the first, second, and third rows, respectively. In the first row, we observe that for small exemplar memory sizes, the performance gap between AHR and the baselines is more significant compared to larger memory sizes across all benchmarks. In the second row, AHR proves to be the least resource-consuming in terms of exemplar memory size and compute (epochs) needed to meet a target performance. In the third row, the performance is reported for various allocated wall-clock times.

**Decoder impact.** Fig. 4 examines decoder sizes of $[0.6, 0.8, 1, 1.2, 1.4]$ and $[1, 1.2, 1.4, 1.6, 1.8]$ million parameters for *CIFAR-100*(10/10) and *miniImageNet*(20/5) benchmarks, respectively. It is surprising how much memory can be saved—and, conversely, how much performance can be improved—with a relatively simple decoder consisting of just three layers of CNNs, which incurs minimal memory and compute overhead. Table 3 shows that the memory requirement of the decoder

Table 3: Performances for fixed memory (both the decoder and exemplars) and compute budgets.

| Strategies | Benchmarks | # Exemplars | Memory | | # Epochs | Wall-Clock Time | Performance |
| | | | Decoder | Exemplar | | | |
|---|---|---|---|---|---|---|---|
| AHR | CIFAR-100(10/10) | 150 (latent) | 1.4M | 4.6M | 50 | 462min | **54.43** ±0.93 |
| | miniImageNet(20/5) | 190 (latent) | 1.8M | 40.54M | 70 | 842min | **48.09** ±0.64 |
| BiC | CIFAR-100(10/10) | 20 (raw) | - | 6M | 60 | 473min | 52.12 ±0.91 |
| | miniImageNet(20/5) | 20 (raw) | - | 42.34M | 80 | 837min | 45.23 ±0.62 |
| IL2M | CIFAR-100(10/10) | 20 (raw) | - | 6M | 60 | 455min | 50.81 ±0.74 |
| | miniImageNet(20/5) | 20 (raw) | - | 42.34M | 80 | 861min | 44.67 ±0.63 |
| EEIL | CIFAR-100(10/10) | 20 (raw) | - | 6M | 60 | 478min | 51.70 ±0.79 |
| | miniImageNet(20/5) | 20 (raw) | - | 42.34M | 80 | 859min | 41.83 ±0.55 |

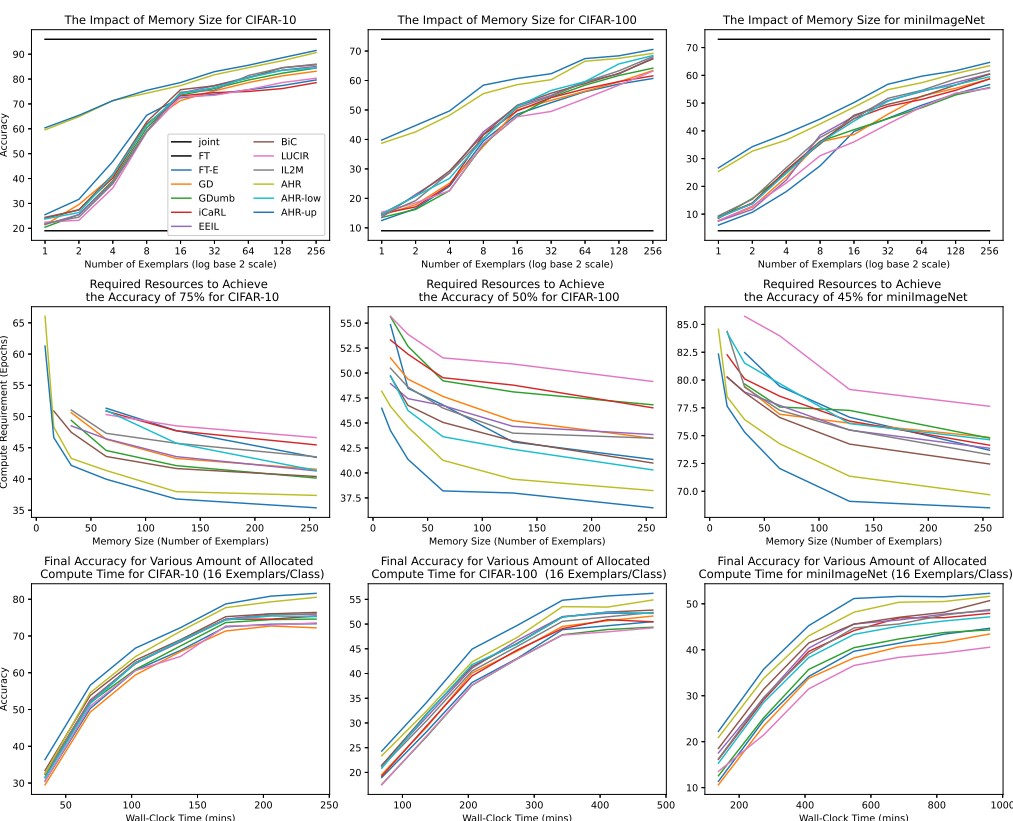

Figure 5: The impact of the memory size (first row). The required resources to achieve a target performance (second row). The achieved performance for a given compute time (third row).

is negligible compared to storing raw exemplars, and that the encoder-decoder architecture of AHR makes it possible to store one order of magnitude more exemplars in the latent space. Overall, Table 3 demonstrates that AHR delivers superior results with the same memory/compute footprints.

## 5 CONCLUSION

Exemplar replay strategies rely on storing raw data, which can be highly memory-consuming, especially since datasets usually require orders of magnitude more memory than models. Meanwhile, generative replay, training a (generative) model for generating the pseudo-data of past tasks, requires far less memory but tends to be less effective. We proposed AHR, a hybrid approach that combines the strengths of both methods, utilizing HAE with RFA for the incremental embedding of new tasks in the latent space. Instead of storing the raw data in the input space, AHR stores them in the latent space of HAE. Our experiments demonstrate the effectiveness of AHR.

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

## A COMPLEXITY ANALYSIS

**Generative replay.** This strategy (Shin et al., 2017) utilizes a *generative model* denoted by $\text{GEN}()$ whose size is not growing with respect to the total number of tasks $I$. $\text{GEN}()$ generates data of the past tasks $\{1, \ldots, T_\ell - 1\}$ to be interleaved with the new task $T_\ell$ to be fed into *discriminative model* $\text{DIS}()$ as well as $\text{GEN}()$ lest they are forgotten. Since neither the size of $\text{GEN}()$ nor $\text{DIS}()$ is growing with respect to the total number of tasks $I$, the memory complexity can be said to be $\mathcal{O}(cte)$. The compute complexity, however, depends on the total amount of data to be fed to both $\text{GEN}()$ and $\text{DIS}()$ which is a function of the number of tasks seen so far if they are not to be forgotten. Hence, the compute complexity is $\mathcal{O}(t)$.

**Generative classifier.** This strategy (Ven et al., 2021) trains a brand new out-of-distribution detector for each new class $c$ (not task) denoted by $\text{OOD}_c()$. Hence, the number of the models grows as new classes arrive indicating that the memory complexity is $\mathcal{O}(t)$. Because this strategy trains a brand new model each time, it does not have to overwrite previous knowledge, and therefore, does not require replaying of the old data. As a result, the amount of data to be fed into each $\text{OOD}_c()$ whenever each model is trained does not grow with the number of classes so far because only the data of class $c$ is fed into $\text{OOD}_c()$. Hence, the compute complexity is $\mathcal{O}(cte)$.

**Exemplar replay.** This strategy (Rebuffi et al., 2017) uses a memory denoted by $\text{MEM}$ that stores dozens of exemplars per class so that each time a new task arrives a representative minibatch of data consisting of all previous tasks is fed into the discriminative model denoted by $\text{DIS}()$. In this strategy, both the memory $\text{MEM}$ has to grow in proportion to the number of tasks, and, the amount of data each time the discriminative model $\text{DIS}()$ must consume has to increase in proportion to the number of tasks seen so far, therefore, memory and compute complexities are $\mathcal{O}(t)$.

**Hybrid replay (ours).** Although learning new tasks requires high-quality data, mitigating forgetting, as it has been reported (Ven et al., 2020), can be effectively accomplished with tolerably lossy data. Leveraging that, our hybrid replay strategy, a combination of generative and exemplar replay, proposes to use an autoencoder consisting of an encoder denoted by $\text{ENC}()$ and a decoder denoted by $\text{DEC}()$. $\text{ENC}()$ serves two goals: (i) it maps the input data to the latent space making it possible to use Euclidean distance for classification, and (ii) it compresses down the input data such that they can be stored in the memory $\text{MEM}$ efficiently. Meanwhile, $\text{DEC}()$ decompresses the data in $\text{MEM}$ each time learning a new task. Since $\text{ENC}()$ compresses down the input data, 10 times at the very least in our experiments, the memory complexity becomes $\mathcal{O}(0.1t)$, whereas the compute complexity is $\mathcal{O}(t)$. Note that the introduced overhead by adding $\text{DEC}()$ is negligible as discussed in the experimental results section.

## B HYPERPARAMETERS AND IMPLEMENTATION DETAILS

We outline the hyperparameters for the 19 CIL strategies including vanilla and joint strategies serving as the lower and upper bounds for image classification on 5 benchmarks as specified in Table 4.

Table 4: Number of latent exemplars for each benchmark for our AHR strategy.

| Dataset | MNIST | SVHN | CIFAR-10 | CIFAR-100 | miniImageNet |
|---|---|---|---|---|---|
| Tasks Configuration | (5/2) | (5/2) | (5/2) | (10/10) | (20/5) |
| # Tasks | 5 | 5 | 5 | 10 | 20 |
| # Classes/Task | 2 | 2 | 2 | 10 | 5 |
| # Classes | 10 | 10 | 10 | 100 | 100 |
| Model | Dense | ResNet32 | ResNet32 | ResNet32 | ResNet32 |
| Learning Rate | 0.001 | 0.001 | 0.001 | 0.001 | 0.001 |
| Momentum | 0.9 | 0.9 | 0.9 | 0.9 | 0.9 |
| # Epochs | 40 | 50 | 50 | 50 | 70 |
| Minibatch Size | 128 | 128 | 128 | 256 | 256 |
| Input Dimensions | $28 \times 28 \times 1$ | $32 \times 32 \times 3$ | $32 \times 32 \times 3$ | $32 \times 32 \times 3$ | $84 \times 84 \times 3$ |
| Input Size | 784 | 3072 | 3072 | 3072 | 21168 |
| Latent Size | 20 | 307 | 307 | 307 | 2117 |
| Compression Ratio | $\approx 40$ | $\approx 10$ | $\approx 10$ | $\approx 10$ | $\approx 10$ |
| # Raw Exemplars | 200 | 200 | 200 | 2000 | 2000 |
| # Latent Exemplars | 8000 | 2000 | 2000 | 20000 | 20000 |

