# OpenReview forum: "Autoencoder-Based Hybrid Replay for Class-Incremental Learning"
_ICLR.cc/2025/Conference — ICLR 2025 Conference Withdrawn Submission_

### Official Review · Reviewer_BFBo · 2024-10-18

**Soundness:** 2
**Presentation:** 1
**Contribution:** 2
**Rating:** 3
**Confidence:** 4

**Summary:**

The paper addresses the problem of Class-Incremental Learning by introducing a Hybrid Autoencoder (HAE). The proposed model is designed for both exemplar replay during incremental training and classification at inference. The autoencoder consists of two components: an encoder, which is trained to minimize the Euclidean distance between the latent representation and the corresponding class centroid, and a decoder, which is trained to minimize the reconstruction error between the input and output images. Both the encoder and decoder are trained with a distillation loss on the previous task to mitigate activation drift.

At the encoder level, the class centroids serve as anchors in the latent space, guiding the latent representations towards their respective classes. These centroids are initialized before training using the Charged Particle System Energy Minimization (CPSEM) method, which ensures that the centroids are well-separated. After training, a nearest-mean classification rule is applied to classify test images based on the proximity of their latent representations to these centroids.

In the post-training phase, the encoder's latent space output is used to populate a replay buffer with latent representations of the current task samples, following a herding strategy. These representations are replayed in the next task by feeding them into the decoder trained on the previous task, which generates the corresponding images. These generated images are then used for training on the new task. Instead of storing images, as is typically done in incremental learning, the proposed method reduces memory requirements by storing only the latent space representations.

The authors compare their approach to several incremental learning methods on benchmarks such as MNIST, Balanced SVHN, CIFAR-10, and miniImageNet. They also analyze the method's resource consumption, evaluate different decoder sizes, and provide an ablation study by comparing its performance when real images are used during the replay phase instead of the generated ones.

**Strengths:**

- The Charged Particle System Energy Minimization (CPSEM) method for initializing class centroids for the encoder training is novel and interesting.
- Several comparisons are carried out in the experimental section, and the method shows good performance on the benchmarks and methodologies used for comparison.
- Detailed analysis on resource consumption is provided.
- The ablation study comparing the proposed AHR with AHR using original images (AHR-lossless) highlights that the quality of the images generated by the encoder is sufficiently good for replay, as AHR with original images achieves similar performance. I appreciated this analysis.

**Weaknesses:**

Overall, I believe the presentation of the paper requires significant improvement. Below are my major concerns regarding the presentation:

- (a) The complexity analysis in the introduction needs to be clarified and expanded. The notation $O(0.1t)$ is incorrect according to the definition of Big-O notation. What does this represent? If the authors intend to convey that memory is saved by storing the latent space representation, I suggest incorporating both the latent space dimension and image size into the complexity analysis. Additionally, the term $e$ in $O(cte)$ is not defined and requires clarification.

 - (b)  The notation throughout the paper is difficult to follow, with multiple indices used unnecessarily (e.g., in Equation 1). This excessive use of notation makes the paper hard to read. I suggest simplifying the notation wherever possible.

- (c) Equation 1 seems to imply that all examples from previous tasks are needed to minimize the reconstruction error. However, I understand that this is not the case—some examples are real images, while others are generated. The replay buffer should be explicitly highlighted to clarify this in the equation.

- (d) The three pseudocode blocks on page 4 make the methodology difficult to follow. Including all three pseudocode blocks on a single page compresses the accompanying methodology description into less than half a page. As a result, some LaTeX formulas in the main text break across two lines, further increasing the difficulty of reading.

 - (e) The organization of the paper should be reconsidered. Given the limited space for submission, dedicating more than two pages to the literature review while allocating just over one page to the methodology does not allow for a proper description of the proposed approach. I suggest moving the extended literature review to the appendix and presenting a more concise version in the main paper.

Regarding the methodology  and experimental section, my major concerns are as follows:

- (f) The introduction of the Charged Particle System Energy Minimization (CPSEM) for initializing class centroids is interesting but requires additional explanation. It is unclear why this type of initialization benefits the autoencoder and how it relates to the Coulomb interaction energy . While I do not expect a full background on the calculus of variations for minimizing energy, more mathematical details—even in the appendix—would be helpful. An analysis of how centroids are distributed in the latent space is required to underline why the proposed strategy is effective. Furthermore, the CCE (class centroid embedding) placement  is explained only through pseudocode, with no accompanying description. At a minimum, the operations performed in Algorithm 2 should be explained in words to provide intuition, especially for readers unfamiliar with the physics-based intuition behind this algorithm.

- (g) Regarding CCE, why is this initialization considered effective? If the goal is to initialize centroids such that the class centroids are distant from each other, why not simply use the K-means algorithm? Alternatively, why not select class centroids as the latent vectors for each class that are the most distant from each other in terms of Euclidean distance, in a similar way as performed with hard negative sampling?

- (h) The usage of the latent space for memory reduction and the decoder for latent replay is not novel. For example, Ayub et al. (ICLR 2021) [1] employ the encoder for storing latent representations and replay these latent representations in subsequent incremental learning steps. When the memory budget is reached, the latent representations are compressed into centroids and covariances. A comparison with their approach is necessary. The storage of latent representations for incremental learning and efficient memory replay with autoencoder is also explored in [2].

- (i) Comparison in Table 2. 1) Some comparisons are unnecessary since the methods the authors compare to perform different tasks. For instance, Prediction Error Based Classification (PEC) [3] is designed for online continual learning (single-pass data), while the authors address the problem of offline incremental learning. It is clear that the harder setting for which PEC is designed results in lower performance compared to the authors' method. The authors should compare their approach only with offline class incremental learning methods under the same conditions. 2) Since the authors' method operates in an offline incremental learning setting, they should compare it with recent exemplar-based class incremental approaches, such as X-DER [4] and MEMO [5]. Additionally, the comparison should consider using a larger and more realistic backbone, such as ResNet-18, which is now commonly evaluated with more parameters and a larger latent space [4][5]. The paper should also evaluate how the method performs with higher-resolution images (e.g., 224x224) on a dataset like ImageNet100 [5].

- (l) In Table 3, the authors report the wall-clock time for training their methods. They state that training takes about 8 hours on CIFAR-100 with ResNet-32, which seems excessive. In FACIL [6], joint training on a not particularly novel GPU requires less time, as only about 400k parameters need to be optimized. What are the timings for joint training with the same epoch budget? An incremental learning method should be more efficient than joint training. Additionally, the authors should specify the device used for the experiments when reporting training time.

[1] A. Ayub and A. Wagner, “{EEC}: Learning to encode and regenerate images for continual learning,” in International
Conference on Learning Representations, 2021.

[2] Caccia, L., Belilovsky, E., Caccia, M. &amp; Pineau, J.. (2020). Online Learned Continual Compression with Adaptive Quantization Modules.  Proceedings of the 37th International Conference on Machine Learning, in  Proceedings of Machine Learning Research.

[3] Michał Zaj ˛ac, Tinne Tuytelaars, and Gido M van de Ven. Prediction error-based classification for
class-incremental learning, in ICLR 2024

[4]Zhou, Da-Wei and Wang, Qi-Wei and Ye, Han-Jia and Zhan, De-Chuan, A Model or 603 Exemplars: Towards Memory-Efficient Class-Incremental Learning, in ICLR 2023

[5] Matteo Boschini, Lorenzo Bonicelli, Pietro Buzzega, Angelo Porrello, Simone Calderara. Class-Incremental Continual Learning into the eXtended DER-verse, in TPAMI 2022


[6] Marc Masana, Xialei Liu, Bartlomiej Twardowski, Mikel Menta, Andrew D. Bagdanov, Joost van de Weijer Class-incremental learning: survey and performance evaluation, TPAMI 2022

**Questions:**

Overall, I believe the paper has significant issues with presentation (as outlined in points [a-e] of the weakness section), which necessitate a major reformatting. Additionally, regarding other concerns, the novelty appears limited to performing classification at the encoder level and the introduction of CPSEM for initializing class centroids. While the latter is a novel contribution, it is neither well-explained nor well-motivated (as noted in points [f-g] of the weakness section). The experimental section, particularly the method comparison, needs refinement by considering recent related work on the proposed approach, utilizing higher-resolution benchmarks, and employing architectures with a larger latent space (as indicated in points [h-i] of the weakness section). Moreover, the details about the training resources should be clarified (point [l] in the weakness section).

I believe the paper has the potential for significant improvement for a future submission. To enhance its novelty, the authors should focus more on the description and proposal of the class centroid initialization, which could provide both theoretical and empirical insights. However, as previously mentioned, these insights are lacking in the current version. Additionally, improving the experimental section would further strengthen the submission.

Considering all the above, I recommend rejecting the current submission. The paper is not yet ready for publication and requires significant revisions. I suggest making these improvements and submitting it to a different venue.

---

### Official Review · Reviewer_R7ia · 2024-10-27

**Soundness:** 3
**Presentation:** 2
**Contribution:** 3
**Rating:** 5
**Confidence:** 4

**Summary:**

The paper introduces Hybrid Autoencoder (HAE) and Autoencoder-Based Hybrid Replay (AHR) strategies to reduce the memory burden for CIL, especially for replay-based approaches. HAE combines both discriminative and generative modeling to handle classification and replay tasks. It employs Charged Particle System Energy Minimization (CPSEM) equations and the Repulsive Force Algorithm (RFA) to manage class separation within its latent space, enabling class identification using Euclidean distance. AHR integrates exemplar and generative replay strategies by storing samples in the latent space, which significantly reduces memory usage. Its decoder is designed to memorize training data, allowing for effective replay without the typical issues of hazy pseudo-data found in other generative approaches. Simulations in various benchmark datasets also validate the hypothesis.

**Strengths:**

1) Easy to follow and addresses the important topic of computational burden for continual learning algorithms
2) The motivation behind the Hybrid replay is well written by contextualizing the current works of literature along with their gaps

**Weaknesses:**

1) Inconsistent notation
: In the explanation of AHR (184) you are referring are T_l but in the   algorithm, it is T_i which is the same for P
                                     : what does the * refer to in algorithm 1?
                                     : What is J_l?
                                    : ^ in explanation and ' is used in Figure 1 are interchangeably used for generated output
                                     : what is \mathcal{T} in Figure 1?
 2) Is there any explanation for how the memory is reduced to 0.1t

 3) It is claimed that the complexity reduces to 10%, but no empirical evidence is provided to validate that hypothesis
4) Evaluation metric: It is unclear to readers, (line 373) does the accuracy represents the accuracy on the only last task after training on all tasks or is average on all previous tasks.
5) How does the number of exemplars decrease over time? You are representing the exemplars in latent space? Does it mean reducing the number of classes?
(Table 3) Why the different numbers of epochs for AHR and others? Please be clear on the size of latent and raw ? is 150 (latent) better than 20 (raw)
6) It would be clearer to the readers if there was some explanation of how CPSEM and RFA create incremental embeddings
7) The main objective of this paper is to reduce the size of exemplars in memory. In the related work section, the authors focus on mainly describing the current replay mechanism without mentioning how the current strategies fall short in reducing the size and its relation to AHR


8) There is no comparative analysis of the work with state-of-the-art replay methods such as:
   I) Rolnick, David, et al. "Experience replay for continual learning." Advances in neural information processing systems 32 (2019).

   II) Buzzega, Pietro, et al. "Dark experience for general continual learning: a strong, simple baseline." Advances in neural information processing systems 33 (2020): 15920-15930.

which makes it challenging to assess the significance of the work in the literature.

### Comments on evaluation:
I am mainly concerned about the accuracies for CIFAR100 and mimiImageNet. There are various works [FeTrIL [1] by Petit, Grégoire, et al., FeCAM [2] by Goswami et al] utilizing ResNet-18/32 achieving higher accuracy of more than 65% even in exemplar-free settings. I wonder how with an exemplar, the model is not able to maintain that accuracy.

[1] Petit, Grégoire, et al. "Fetril: Feature translation for exemplar-free class-incremental learning." Proceedings of the IEEE/CVF winter conference on applications of computer vision. 2023.

[2] Goswami, Dipam, et al. "Fecam: Exploiting the heterogeneity of class distributions in exemplar-free continual learning." Advances in Neural Information Processing Systems 36 (2024).

**Questions:**

please refer to weakness section

---

### Official Review · Reviewer_ykiH · 2024-11-02

**Soundness:** 2
**Presentation:** 1
**Contribution:** 3
**Rating:** 5
**Confidence:** 1

**Summary:**

This paper proposes an autoencoder-based hybrid replay (AHR) strategy that leverages our new hybrid autoencoder (HAE) to function as
a compressor to alleviate the requirement for large memory, achieving O(0.1t) at the worst case with the computing complexity of O(t) while accomplishing state-of-the-art performance.

**Strengths:**

1. This paper is well organized.

2. I appreciate the extensive experiments.

3. The idea of modeling the energy dynamics within the system akin to charged particles is interesting.

**Weaknesses:**

1. The writing of Section 2 ("OUR STRATEGY: AUTOENCODER-BASED HYBRID REPLAY (AHR)") is confusing. Please outline the motivations of each step and explain why it makes sense.

2. The technique contribution is fuzzy. I want to know what technique used in this paper and what technical challenge or a novel idea in the technique of proposed method.

**Questions:**

I have read this paper carefully. Unfortunately, this paper is totally out of my research area. Therefore, I cannot capture the brilliance of this paper.

---

### Official Review · Reviewer_zWHK · 2024-11-03

**Soundness:** 3
**Presentation:** 3
**Contribution:** 2
**Rating:** 5
**Confidence:** 4

**Summary:**

The paper presents a novel approach to CIL called Autoencoder-based Hybrid Replay (AHR). This method combines exemplar and generative replay techniques to address key challenges in CIL, such as task confusion and catastrophic forgetting (CF). The hybrid autoencoder (HAE) serves as both a discriminative and generative model, storing data in a compressed latent space with minimal memory (O(0.1t)) compared to traditional exemplar methods (O(t)). The use of charged particle system energy minimization (CPSEM) and a repulsive force algorithm (RFA) aids in optimal placement of class centroids in the latent space. The experimental results indicate that AHR consistently outperforms existing baselines across five benchmarks.

**Strengths:**

+ The combination of generative and exemplar replay in a single system that minimizes memory while maintaining high performance is novel.
+ The proposed method achieves a significant reduction in memory requirements (O(0.1t)), which is crucial for scalability in CIL.
+ Comprehensive experiments across multiple benchmarks and comparisons with state-of-the-art (SOTA) methods demonstrate the robustness of AHR.

**Weaknesses:**

-The paper lacks exploration of real-world applications or more complex, dynamic scenarios beyond standard benchmarks.
-Performance could be impacted if the autoencoder's compression and reconstruction capabilities are not well-optimized.
- While memory reduction is emphasized, the impact of this method on significantly larger-scale datasets or more diverse data distributions is not detailed.

**Questions:**

- What are the specific challenges encountered when integrating CPSEM and RFA into existing architectures?
- Have you considered applying AHR to non-vision data, such as text or audio?

---

### Note · Authors · 2024-11-15

I have read and agree with the venue's withdrawal policy on behalf of myself and my co-authors.